# Outpatient Assessment of Mechanical Load, Heat Strain and Dehydration as Causes of Transitional Acute Kidney Injury in Endurance Trail Runners

**DOI:** 10.3390/ijerph181910217

**Published:** 2021-09-28

**Authors:** Daniel Rojas-Valverde, Ismael Martínez-Guardado, Braulio Sánchez-Ureña, Rafael Timón, Volker Scheer, José Pino-Ortega, Guillermo Olcina

**Affiliations:** 1Centro de Investigación y Diagnóstico en Salud y Deporte (CIDISAD), Escuela Ciencias del Movimiento Humano y Calidad de Vida (CIEMHCAVI), Universidad Nacional, Heredia 86-3000, Costa Rica; 2Clínica de Lesiones Deportivas (Rehab & Readapt), Escuela Ciencias del Movimiento Humano y Calidad de Vida (CIEMHCAVI), Universidad Nacional de Costa Rica, Heredia 86-3000, Costa Rica; 3Grupo Avances en Entrenamiento Deportivo y Acondicionamiento Físico (GAEDAF), Facultad Ciencias del Deporte, Universidad de Extremadura, 10003 Cáceres, Spain; rtimon@unex.es (R.T.); golcina@unex.es (G.O.); 4Faculty of Life and Natural Sciences, University of Nebrija, 28015 Madrid, Spain; 5Programa de Ciencias del Ejercicio y la Salud, Escuela Ciencias del Movimiento Humano y Calidad de Vida, Universidad Nacional, Heredia 86-3000, Costa Rica; brau09@hotmail.com; 6Ultra Sports Science Foundation, 69310 Pierre-Bénite, France; volkerscheer@yahoo.com; 7Biovetmed & Sportsci Research Group, University of Murcia, 30720 San Javier, Spain; josepinoortega@um.es

**Keywords:** kidney failure, AKI, health, biomarkers, strenuous exercise, mountain running, kidney function, off-road running

## Abstract

Background: This study aimed to globally assess heat strain, dehydration, and mechanical load as acute kidney injury (AKI) indicators in amateur endurance trail athletes during a 35.3 km run. Methods: Thirty amateur experienced trail runners completed an endurance trail run (total positive ascend 1815 m). The following assessments were performed at four measurement time points (pre-, during, immediately post [-post_0h_], and after 24 h of the finish of the run [-post_24h_]): serum test (creatinine, blood ureic nitrogen, albumin, creatine kinase, blood ureic nitrogen: creatinine ratio, creatinine clearance, and glomerular filtration rate), mechanical load (impacts and Player Load), heat strain and dehydration (hematocrit, urine solids, body weight and urine specific gravity), pain and exertion perception (rate of perceived exertion, lumbar and bipodal, and one-leg squat pain), and urinalysis (pH, protein, glucose, erythrocytes, and urine specific gravity). Results: There were pre vs. post_0h_ changes in all serum biomarkers (*F* = 5.4–34.45, *p* < 0.01). The change in these biomarkers correlated with an increase in mechanical load indicators (*r* = 0.47–59, *p* < 0.05). A total of 40% and 23.4% of participants presented proteinuria and hematuria, respectively. Pain and perceived exertion increased significantly due to effort made during the endurance trail running (*F* = 4.2–176.4, *p* < 0.01). Conclusions: Endurance trail running may lead to an increase in blood and urine indicators of transitional AKI. The difference in blood and urine markers was significantly related to the mechanical load during running, suggesting potential kidney overload and cumulative mechanical load.

## 1. Introduction

There are activities of increasing popularity in endurance sports, such as running, cycling, triathlon, and open water swimming. This widespread endurance sports practice responds to the economic prosperity of some populations, relatively inexpensive travel cost, and the relatively affordable and user-friendly required equipment for its practice [1]. Endurance exercising usually requires great effort and takes the body to its physiological, cognitive, and physical limits. The cumulative moderate to high-intensity actions over a prolonged period may lead to fatigue, temporal disfunction, injury, or even death in conjunction with other potential risks [2,3]. In this sense, trail running is one of the most physically demanding sports due to the difficulty and duration of the events, terrain characteristics, weather variation, and slope changes [3,4,5,6]. These characteristics require several concentric and eccentric muscle actions, which have proven highly tiring at the neuromuscular level [4,7].

Muscle damage is a common consequence of trail running [3]. Still, recently, there has been an increase in the concern for trail running participants’ well-being due to the prevalence of some adverse health conditions at the cardiovascular, immunological, hepatic, and renal levels [3,8,9,10]. Specifically, at the renal level, physical stress is a factor that contributes to the transitory decrease of renal function [3,11]. This short-term kidney issue is clinically characterized by a rise of nitrogen waste products in the bloodstream in response to mechanical and thermal muscle injury and future inflammation responses that could compromise the balance of fluids in the body [2]. This condition is known as acute kidney injury (AKI), also called acute renal failure, which in most cases is reversible and asymptomatic [8,11].

There are several methods to quantify renal function. It is commonly assessed by chemical waste molecules generated from muscle metabolism. During exercise, due to muscle damage, the protein-waste products released to the bloodstream could overload the kidney. Among these biochemical markers of kidney damage are serum creatine kinase (sCK), serum creatinine (sCr), serum blood ureic nitrogen (sBUN), serum albumin (ALB), and recently through Cystatin-C (Cyst-C) [9,12]. Other indicators less frequently reported are neutrophil gelatinase-associated lipocalin (NGAL) and kidney injury molecule 1 (KIM-1) [13]. From these well-known AKI biomarkers, some other variables could be estimated, such as BUN-to-Cr ratio (BUN/Cr), glomerular filtration rate (eGFR), albumin-to-creatinine ratio (ALB/Cr), and creatinine clearance [14,15]. The clinical diagnosis of AKI has been established based mainly on sCr levels criteria [12]. Nowadays, alternative subclinical and functional measures have been proposed, but there is no scientific consensus [13], and no range values for sports have been established [3].

AKI is understood as a multi etiology condition. There are three factors proposed as enhancers of AKI development in sports: high mechanical load, heat strain, and dehydration [3,16]. Recent research has suggested that high physical load is the common denominator of AKI progress [11,17,18]. AKI has been reported among endurance athletes and other populations that usually perform high physical loads during prolonged periods under hot and humid conditions [11,19]. Heat and dehydration would enhance possible damage and exposure, and partially explain mechanical kidney damage [11].

AKI has been studied in populations with the abovementioned conditions, such as athletes, agricultural workers, firefighters, and builders [16,20]. Although this evidence exists, clarification is needed as to how these three factors contribute to AKI development under isolated conditions and in outpatient settings [2,3]. There has been no consensus on the long-term consequences of subsequent AKI events leading to chronic renal disease [10,21]. Still, some hypotheses considering renal scarring and maladaptive repair due to cumulative AKI events could better understand this link between acute and long-term renal issues [22]. For this reason, recent studies have tried to use gold-standard measurement protocols in outpatient settings that provide valuable information for a better understanding of this phenomenon [11], but more evidence is required.

The prevalence and severity of AKI episodes among endurance athletes have been debated without reaching a consensus. Some authors point out that AKI should be considered a health problem or issue [23], and others consider it a common effect of physical exercise [2,8]. Highly-demanded sports, such as running, cycling, and triathlon, have high load components understood as high volume and high intensity that may contribute to the potential development of a renal condition [11]. Due to the lack of evidence that addresses the three main causal factors of AKI in endurance sports, there is a need to clarify specific gray points on the understanding of this condition and to determine how each of these factors influence separately and wholey the development of AKI [3]. A global assessment of the AKI phenomenon is indispensable. This study aimed to globally assess heat strain, dehydration, and mechanical load as acute kidney injury (AKI) indicators in amateur endurance trail athletes during a 35.3 km run.

## 2. Materials and Methods

### 2.1. Participants

Thirty male amateur trail runners took part in this study (age 39.5 ± 9.23 years, weight 71.26 ± 11.17 kg, height 171.65 ± 8.69 cm). Participants were selected from different trail running clubs. Participants were required to be >18 years old, experienced (5.69 ± 2.77 years), trained (9.02 ± 3.57 h/week), and heat acclimatized (sleep and train in similar study’s altitude and weather) endurance runners. Participants reported no neuromuscular or metabolic disturbance or injury at least six months before the study.

The experimental protocol was approved by the Institutional Review Board of the National University of Costa Rica (Reg. Code UNA-CECUNA-2019-P005) and the University of Extremadura (Reg. Code 139/2020). All the participants were informed of the details of the experiment procedures, the associated risks and discomforts, and their benefits and rights. Each subject gave written informed consent, according to the criteria of the Declaration of Helsinki, regarding biomedical research involving human subjects (18th Medical Assembly, 1964, revised in 2013 in Fortaleza).

### 2.2. Study Design

Participants were measured in four different time points (pre-, during, -post_0h,_ and -post_24h_) (Figure 1). All participants were asked to trail run a total distance of 35.3 km (cumulative positive ascend of 1815 m, lowest and highest altitude = 906 and 1178 m.a.s.l.). The thermal stress index was registered at the start line of the trail run. The mean thermal index was 24.31 ± 1.6 °C (temperature: 25.52 ± 1.98 °C and humidity 79.25 ± 7.45%) according to the WetBulb-Globe Temperature (WBGT). The participants spent 294.14 ± 59.34 min to finish the three loops.

As shown in Figure 1, serum and urine samples and heat strain and hydration variables were assessed ~15 min before and after the trail run. A follow-up was performed 24 h after (individual visits). The participants were allowed to carry their hydration throughout the event, so their liquid and food intake were established *ad libitum* as typical in trail running races. Perceived pain and exertion data was collected ~10 min before, after, and during the run. Mechanical load variables were monitored all over the trail run using wearable devices.

### 2.3. Materials and Procedures

For serum test analysis, blood was extracted *in-situ* from the antecubital vein using a 5 mL blood collection sterile tube (BD Vacutainer^®^, Franklin Lakes, NJ, USA) that contained spray-coated silica particles activator and gel polymer to facilitate serum separation during centrifugation (10 min at 2000× *g* relative centrifugal force). Centrifugation was performed using a tube centrifuge (PLC-01, Gemmy Industrial Corp., Taipei, Taiwan).

Blood samples were stored on ice in a special cooler (45QW Elite, Pelican^TM^, California, USA) until serum samples were frozen at −20 °C (~5 h after blood extraction). Samples analyses and processing were performed 24 h after data collection in an isolated and temperature-controlled laboratory using an automatic biochemical analyzer (BS-200E, Mindray, Beijing, China) by photometry method. All procedures were executed under relevant protocols for the handling and disposal of biological materials, according to the manufacturer’s instructions for the equipment and reagents used.

The variables analyzed were serum creatinine (sCr, mg/dL), creatine kinase (sCK, IU/L), ureic nitrogen (sBUN, mg/dL), and albumin (sALB, IU/L). The following variables were also estimated: eGFR (mL/min/1.73 m^2^), Cr_Clearance_ (mL/min) and sBUN/sCr ratio. Cr_Clearance_ was predicted from sCr using the Cockcroft and Gault [14] estimation equation:(1)$${\mathrm{Cr}}_{\mathrm{Clearance}}=\left(140-\mathrm{age}\right)*\;\frac{\mathrm{body}\;\mathrm{mass}\;\left(\mathrm{kg}\right)}{\mathrm{sCr}\;\left(\mathrm{mg}/\mathrm{dL}\right)}*72$$

eGFR was calculated using CKD-EPI [15] sCr-based formula for male and white population as follow: (2)$$\mathrm{eGFR}=141*\left(\frac{\mathrm{sCr}\;\left(\mathrm{mg}/\mathrm{dL}\right)}{0.9}-1.209\right)*0.993*\mathrm{age}$$

Urine samples were collected in-situ in a 30 mL polypropylene sterile urine container (Nipro Medical Corp., Osaka, Japan) and analyzed with highly sensitive and accurate dipsticks for urine screening (Combur_10_Test M, Roche, Mannheim, Germany), previously used in distance running settings [24]. Urine dipsticks were examined immediately after collection by two different observers simultaneously using the color scale reference provided by the manufacturer. In case of disagreement between observers, a consensus was obtained with the opinion of a third observer. The following parameters were screened: pH (acidity and basicity), protein, glucose, erythrocytes, and urine specific gravity (USG). There were no reported urination problems or difficulties neither before nor after the study. Urine solids were assessed, and USG was confirmed and double-checked with a digitally valid [25] handheld refractometer (Palm Abbe^TM^, Misco, OH, USA). USG results were classified following the hydration status ranges: well-hydrated <1.01, minimal dehydration 1.01–1.02, significant dehydration 1.02–1.03, and severe dehydration >1.03 [26]. The refractometer was previously cleaned with distilled water and calibrated.

Additionally, a sample of capillary blood was extracted from the right index finger, using a Na-heparinized capillary tube (80 IU/mL) (Marienfeld, Lauda-Königshofen, Germany) to assess hematocrit (Htc). The capillary micro-hematocrit tubes were centrifuged (KHY-400, Gemmy Industrial Corp., Taipei, Taiwan), and hematocrit values were evaluated using a special reader (Gemmy Industrial Corp., Taipei, Taiwan).

Participant’s body mass (kg) was assessed semi-nude (underwear only) immediately pre and post-event using a digital balance (BC554, Tanita, Arlington Heights, IL, USA) in an isolated tent. Percentage body weight change was categorized as follow: well-hydrated +1% to −1%, minimal dehydration −1% to −3%, significant dehydration −3% to −5% and serious dehydration >5% [26]. Liquid and food intake were set ad libitum and were not controlled to maintain organic trail run conditions.

During the whole event, variations of the WetBulb-Globe Temperature (WBGT) were registered with a heat stress monitor (QUESTemp^TM^ 36, 3M^TM^, Saint Paul, MN, USA) every 15 min. Equipment was calibrated and mounted in a stable tripod at the start and finish of the event.

Mechanical load variables were assessed using six inertial measurement units (WIMU PRO^TM^, RealTrack Systems, Almería, Spain) attached to different anatomical spots (one IMU at thoracic 2nd–4th [T_2_–T_4_], one IMU at lumbar 1st–3rd [L_1_–L_3_]; two IMU at right [VL_right_] and left [VL_left_] vastus lateralis muscle bellies and two IMU 3 cm cephalic to right [MP_right_] and left [MP_left_] malleolus peroneus). IMUs were mounted using special spandex dark suit to avoid non-desired movement or shaking during running, as reported in previous studies [11].

The IMUs reliability during multidevice assessment had been tested [27]. Before the study was performed, all devices were calibrated following published protocols [27]. All IMU’s were turned on 30 min before the matches started to reach the optimal internal temperature of the device (~32 °C). The sampling frequency of accelerometry-based external load indicators was set at 100 Hz. Each participant’s devices were synchronized in time before analysis, and data filtration processes were applied (at chip-level and data processing algorithms) before the ¨raw data¨ was available for researchers. Default by the manufacturer made this filtering, and data processing was performed based on the redundancy principle and fusion of the incorporated sensors (accelerometer, magnetometer, and gyroscope). Total variables´ data were extracted from IMU 15 min after the participants finished using special software.

The analyzed variables were selected considering similar studies investigating mechanical load and AKI in endurance events [11]. The extracted variables were: Player Load per min (AU, PL/min) and impacts (Impacts_total_, n/min) per body segment. Player Load is defined as the vector sum of accelerometer data points in the three axes of movement [28], representing the sum of the magnitudes of the movement. The impacts are defined as a short duration jerk, shock, or impact, expressed in gravitational forces (*g*-forces) where 1 *g* = 9.81 m/s [18,29].

Perceived pain and rate of perceived exertion (RPE) were measured using a 0 to 10 point visual analog scale of pain (VAS-PAIN); cero is understood as no pain, and ten is an extreme pain [30]. This variable was assessed through three different tasks as follows: lumbar pain, asking participants for lumbar pain (pain in the lower back); squat pain, perceived pain after a 2 s-90° knee squat to evaluate knee and hip flexors and extensor pain during concentric contraction; and one-leg squat pain, same as the second task (but only with reported dominant leg). The RPE during the event was assessed using the modified Borg Scale, and it was evaluated in each of the control points (each 11.46 km).

### 2.4. Statistical Analysis

Description of variables was reported using mean, standard deviation, and lower and upper limits. The normality of the data was confirmed using the Shapiro–Wilk test. Differences between variables were explored using one-way analysis of variance (serum test, specific gravity, heat strain, and hydration); the exploration of the measurement timepoint differences was made using Bonferroni Post Hoch when required. Hematocrit and body weight data were explored using paired *t*-test analysis. The magnitude of the differences (effect size) was qualitatively interpreted using partial omega squared (*ω_p_²*) as follows: >0.01 small; >0.06 moderate and >0.14 large and using Cohen *d* (*d*) as follows: <0.2 trivial; 0.2–0.49 small; 0.5–0.79 moderate and >0.8 large [31] when corresponded.

McNemar non-parametric test was used to explore the possible change in proportion for the paired data of urinalysis. In those observed cases, the intersection frequency value was <5, and the binomial test was performed. The data of pH, glucosuria, proteinuria, hematuria, and USG were paired by measurement timepoints using a 2 × 2 contingency table. A Pearson Correlation matrix was performed and graphed to explore the correlation between serum markers’ change percentage (Δ%) and mechanical load variables.

Alpha was prior set as *p* < 0.05. Data analysis was performed using the Statistical Package for the Social Sciences (SPSS, IBM, SPSS Statistics, v.22.0, Chicago, IL, USA).

## 3. Results

When compared pre and post_0h_ and pre and post_24h_ measurement time points, the levels of sCr, sCK, sBUN, sALB, USol, and sBUN/sCr ratio significantly increased (*p* < 0.01). Moreover, a significant decrease in Cr_Clearance_, eGFR, and BW was presented after the trail run (*p* < 0.01). Some variables, such as sALB, sCK, sCr, eGFR, USG, USol, Cr_Clearance,_ and sBUN/sCr ratio also recovered the baseline values as evidence in post_24h_ assessment (see Table 1).

Figure 2 shows the number of out-of-range cases based on clinical criteria for each variable, following previous studies’ references as stated in the method section. In this sense, post_0h_ presented the higher out of range cases for AKI-related variables, such as sCr (63.0%, 19/30), sBUN (40.0%, 12/30), sALB (13.0%, 4/30), sCK (73.0%, 22/30), Cr_Clearance_ (90.0%, 27/30), and eGFR (30.0%, 10/30).

A significant change in proteinuria was found, presented as an increase in 40.0% of cases with 1+ or higher (pre vs. post_0h_). The baseline levels were recovered after 24 h in 40.0% of these cases. Hematuria cases increased by 23.4% (pre vs. post_0h_) and then decreased by 20.0% (post_0h_ vs. post_24h_). Finally, cases of pH ≤ 5 decrease by 20.0% (pre vs. post_24h_) (see Table 2).

Perceived pain and physical perceived exertion changed throughout the running event. RPE, squat pain, one-leg squat pain increased (*p* < 0.01) gradually between pre, 1st, and 2nd lap. Lumbar pain increased during the race, but no difference was found between the 1st and 2nd lap (see Table 3).

The percentage of change (Δ%) of sCK correlated significantly with PL MP_left_ (*r* = 0.59, *p* < 0.01), PL VL_left_ (*r* = 0.53, *p* = 0.02) and PL T_2_–T_4_ (*r* = 0.49, *p* = 0.04). Besides, the Δ% of sBUN correlated with PL MP_left_ (*r* = 0.53, *p* = 0.02) and PL VL_left_ (*r* = 0.47, *p* = 0.04). Finally, the Δ% of sALB correlated with PL MP_left_ (*r* = 0.6, *p* < 0.01), PL VL_left_ (*r* = 0.47, *p* = 0.04), PL L_1_–L_3_ (*r* = 0.52, *p* = 0.03) and PL T_2_–T_4_ (*r* = 0.49, *p* = 0.04) (see Figure 3).

## 4. Discussion

The purpose of this study was to globally assess heat strain, dehydration, and mechanical load variables as AKI indicators in amateur endurance trail athletes during a 35.3 km run. A significant change between pre vs. post_0h_ assessments in all serum biomarkers (sCr, sCK, sBUN, sALB, Cr_Clearance_) was found, and its percentage of change correlated with mechanical load indicators (Player Load as accelerometry-based load index). A total of 40.0% and 23.4% of cases presented concomitant proteinuria and hematuria, respectively, due to the trail running. Pain and perceived exertion changed significantly after running. Considering selected reference criteria, there was an incidence of 63.0% (19/30) out of range cases in sCr, 40.0% (12/30) in sBUN, 13.0% (4/30) in sALB, 90.0% in Cr_Clearance_ (27/30), and 30.0% (10/30) in sGFR after endurance running. These out-of-range cases in the AKI biomarker indicators returned to baseline after 24 h, suggesting transitional AKI. The outcomes of this study should be analyzed based on two considerations, mean change between timepoint measures and increase or decrease in cases out of range based on clinical and technical criteria. The isolated analysis of statistical differences could lead to misunderstanding the results and the effect of endurance trail running on AKI.

As found in this study (*F* = 5.4–34.5, *p* < 0.01), previous evidence has suggested an increase in biomarkers related to AKI after endurance running events [3]. Distance running events are considered one of the most physically demanding sports [3,22]. Consequently, relative common increases in sCK values during these kinds of events may indicate a high muscle damage rate due to the release of sarcoplasmic proteins into the bloodstream. Damage and disintegration of muscle fibers are common during strenuous physical exertion [34,35]. The structural and functional damage suffered during running could be exacerbated due to the repetitive concentric–eccentric muscle contractions when running uphill and downhill as in endurance trail events [4,6]. This kind of effort requires greater impact absorption and higher metabolic rate [4,6,11].

The physical effort and biomechanical characteristics of trail running could explain the relation found (*r* = 0.47–59, *p* < 0.05) between mechanical load indicators (e.g., impacts and Player Load) registered in different body segments and some AKI biomarkers (sBUN and sALB) [7,11]. Indeed, the mechanical load could increase some kidney functional and subclinical injury biomarkers as sCr, sBUN, sALB [11,36,37], as well as some other serum and urinary kidney function indicators as Cr_Clearance_, eGFR, hematuria, and proteinuria; similar changes to those found in this study [8,19]. In this sense, previous evidence has suggested muscle and kidney mechanical trauma in non-contact sports as endurance running, explained by the high magnitude and number of forces involved during trail running [11,38]. Running downhill, sudden changes of direction, jumps, stops, and other high-intensity actions during trail running could partially mediate the renal dysfunction, considering potential kidney shacking [11,38].

The data presented in this study reinforces the hypothesis of potential cumulative kidney mechanical trauma [11,39], by presenting relationships between the change of sBUN and sALB with the mechanical load of the lumbar region (accelerometer in L_1_-L_3_). It has been found that, due to higher intensity and higher external load presented in shorter and faster endurance events, runners usually have a more significant impact on kidney health than more extended events [8,9,11,37]. Running speed seems to be a crucial factor in developing the temporary reduction in kidney function due to strenuous exercising [9,37].

Additionally, based on selected AKI diagnosis criteria, the change in sCr may suggest a risk of AKI after running endurance trail running, but with a return to baseline values after 24 h, as evidenced in previous studies [3,37]. The increased sCr, sBUN, sCK, sALB, eGFR, Cr_Clearance,_ and sBUN/sCr ratio, with values over the upper reference limits supports this finding [12,40]. Moreover, the increase in urine hematuria and proteinuria reinforces the evidence that AKI is a transitional condition that could be relatively common after strenuous efforts [22,24]. Endurance trail running may induce a decrease of renal perfusion and a disruption of physiological mechanisms that may maintain glomerular filtration during exercise. This effect is corrected until regular kidney function is achieved, apparently after 24 h [8,37].

In addition to the mechanical load, it has been reported that heat strain and dehydration during these kinds of events can boost the prevalence of AKI in endurance running as it has been reported in other populations [16,19,41], due to the decreased blood plasma and consequent reduction in renal blood flow [19]. In the present study, despite relatively favorable weather conditions (thermal index 24.31 °C) compared to other reported studies during endurance running, and considering the out-of-range cases, 66.7% of total participants presented significant dehydration [26] immediately after the race. Still, a 46.7% of the participants remained dehydrated 24 h after the event. It should also be considered that there were *ad libitum* food availability and fluid intake during the race. However, although dehydration could typically contribute to AKI, it seems to have a mild impact, as previously reported [42].

Anecdotally, the change (F = 4.2–176.4, *p* < 0.01) in perceived pain and effort values during the race can be a complementary analysis to the biochemical, mechanical, and thermal load assessments to monitor endurance athletes throughout an event to prevent the appearance of AKI. Lumbar pain and its relationship with AKI biomarkers should be explored in future studies.

### Limitations

While the results of this study have provided information regarding the influence of heat strain, mechanical load and dehydration in the development of AKI, some limitations to the study must be acknowledged. One of the main limitations of this study concerns the relatively small sample; it would be interesting to extend this research to include more participants. Secondly, we must bear in mind that these findings can only be extrapolated to male amateur well-trained runners. Future exploration may analyze AKI in top-elite runners, master athletes, and women populations. Furthermore, the influence of physical fitness and experience on the prevalence and development of AKI should be explored among runners.

Some contextual factors as hydration, food intake, and supplements during running should be controlled in future studies. Moreover, despite its limited access, it may be interesting to assess some novel AKI indicators as Cyst-C, NGAL, and KIM-1 as subclinical AKI markers. It is fundamental to develop a cohort follow-up to confirm the potentiality of cumulative AKI events leading to CKD.

## 5. Conclusions

Endurance running could lead to some temporal dysfunctions, such as AKI. These complications were reflected by the increase of some serum changes as sCr, sCK, sBUN, sALB, eGFR, Cr_Clearance_, and sBUN/sCr ratio. This rising pattern was found in proteinuria and hematuria cases Post_0h_. These increased serum and urine values returned to baseline after 24 h, which may suggest transitional AKI with no additional complications. Perceptual variables as effort and pain increased gradually throughout the race may be related to muscle damage and fatigue. Higher mechanical load (Player Load) correlated significantly with the percentage of change (Pre vs. Post_0h_), sCK, sBUN, and sALB; this was evidence for mechanical muscle damage and strengthened the hypothesis of kidney trauma due to cumulative low to moderate contusions during running [11].

## 6. Practical Applications

Considering endurance trail running could increase some blood and urine samples related to transitory AKI, some factors such as dehydration, heat strain, and mechanical load should be monitored during training and competition to prevent potential future damage.

Managing fluid intake and restoring electrolytes before, during, and after endurance events may reduce the number or lessen the severity of AKI cases. Avoiding repeated endurance events without the required rest and recovery between exhaustive efforts could be protective against AKI.

Although there is insufficient evidence to relate AKI events with chronic kidney diseases, medical staff and trainers may monitor renal health of long-distance runners frequently to detect any change in renal function that could affect runners’ health.

## Figures and Tables

**Figure 1 ijerph-18-10217-f001:**
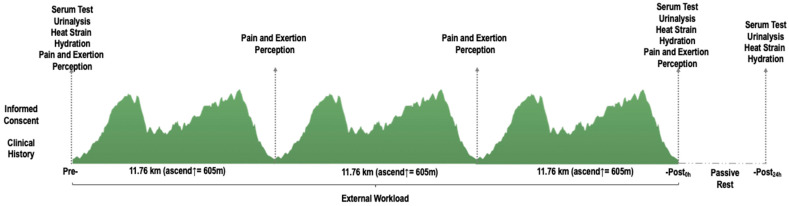
Schematic design of study variables with measurement time and trail altimetry.

**Figure 2 ijerph-18-10217-f002:**
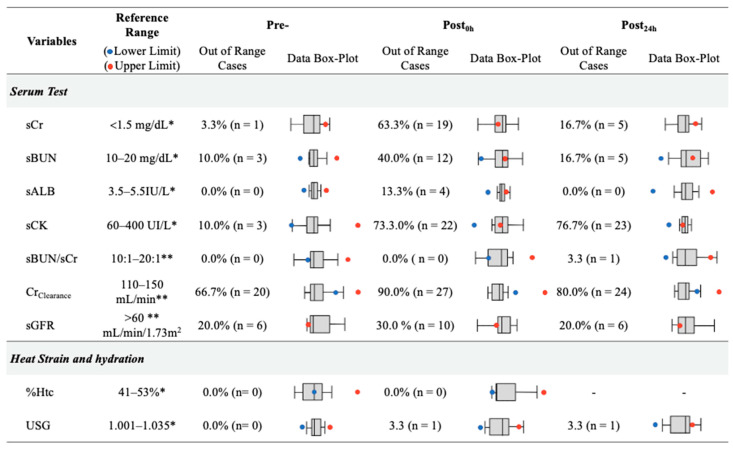
Cases out of reference ranges in serum, heat strain and hydration variables by measurement timepoints. Reference Ranges: * [32], ** [33]. sCr = creatinine, sCK = creatine kinase sBUN = blood ureic nitrogen, sALB = albumin, eGFR, estimated glomerular filtration rate, Cr_Clearance_ = creatinine clearance, USG = urine specific gravity, Htc = hematocrit.

**Figure 3 ijerph-18-10217-f003:**
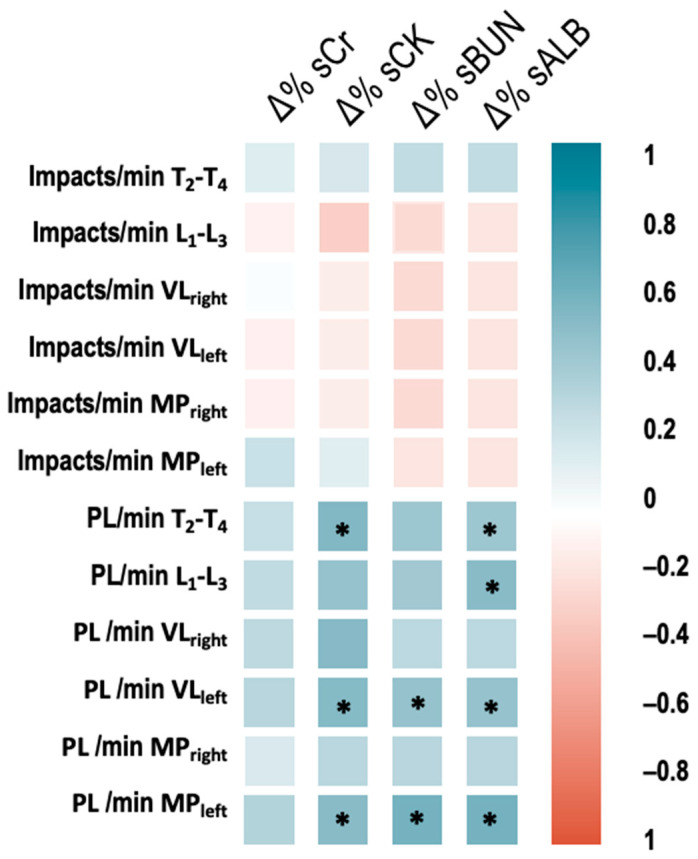
Correlation between percentage of change (pre vs. post0h) of serum values and accelerometric load (impacts and Player Load). * significant correlation *p* < 0.05. PL = player load, T_2_–T_4_ = thoracic 2nd–4th, L_1_–L_3_ = lumbar 1st–3rd; VLright and VLleft = vastus lateralis and MPright and MPleft = malleolus peroneus.

**Table 1 ijerph-18-10217-t001:** Pre-, -Post changes in serum, heat strain, and hydration variables.

Variable	Pre-	-Post_0h_	-Post_24h_	*F/t* Value	*p*-Value	*ω_p_*^2^/*d* Rating
*Serum test*						
sCr (mg/dL)	1.2 ± 0.3 (1.1 to 1.3)	1.6 ± 0.3 * (1.5 to 1.8)	1.3 ± 0.3 ^†^ (1.2 to 1.4)	34.5	<0.01	0.5 *large*
sCK (IU/L)	187.3 ± 182.2 (106.5 to 268.0)	528.0 ± 345.2 * (374.9 to 680.9)	677.0 ± 534.2 * (440.1 to 913.8)	16.3	<0.01	0.3 *large*
sBUN (mg/dL)	14.5 ± 4.1 (12.7 to 16.3)	19.4 ± 4.6 * (17.3 to 21.4)	18.9 ± 4.2 * (17.0 to 20.8)	26.9	<0.01	0.5 *large*
sALB (IU/L)	4.5 ± 0.7 (4.2 to 4.9)	5.1 ± 0.8 * (4.7 to 5.4)	4.7 ± 0.2 (4.6 to 4.8)	7.8	<0.01	0.2 *large*
eGFR (mL/min/1.73m^2^)	80.5 ± 24.7 (69.5 to 91.4)	69.9 ± 18.5 * (61.7 to 78.1)	74.7 ± 21.3 (65.3 to 84.1)	5.4	<0.01	0.1 *moderate*
Cr_Clearance_ (mL/min)	87.4 ± 30.0 (74.1 to 100.7)	60.9 ± 25.9 * (49.4 to 72.4)	78.1 ± 30.8 *^†^ (64.4 to 91.7)	43.8	<0.01	0.6 *large*
sBUN/sCr ratio	12.2 ± 3.3 (10.8 to 13.7)	12.0 ± 2.7 (10.8 to 13.3)	15.3 ± 4.0 *^†^ (13.5 to 17.1)	15.7	<0.01	0.3 *large*
*Heat Strain and Hydration*						
USG	1.02 ± 0.02 (1.0 to 1.0)	1.02 ± 0.01 (1.0 to 1.0)	1.02 ± 0.01 (1.0 to 1.0)	0.8	0.5	0 *trivial*
USol	3.7 ± 2 (2.6 to 4.8)	5.4 ± 2 * (4.3 to 6.5)	4.7 ± 2.3 (3.5 to 6.0)	4.5	0.01	0.1 *moderate*
Htc (%)	41.2 ± 2.9 (40 to 42.4)	42.4 ± 3.7 (41 to 43.8)	-	−1.4	0.2	0.3 *small*
BW (kg)	71.7 ± 10.8 (67.6 to 75.8)	67.8 ± 15.8 * (61.7 to 73.9)	-	2.3	0.03	0.5 *moderate*

Data was presented in mean ± standard deviation and 95% upper and lower limits. Significant differences (*p* < 0.01) with * Pre- and ^†^ Post_0h_. sCr = creatinine, sCK= creatine kinase sBUN = blood ureic nitrogen, sALB = albumin, eGFR, estimated glomerular filtration rate, Cr_Clearance_ = creatinine clearance, USG = urine specific gravity, USol = urine solids, Htc = hematocrit, BW = body weight.

**Table 2 ijerph-18-10217-t002:** Distribution and pattern of change of urine dipstick readings of pH, proteinuria, glucosuria, hematuria, and USG.

Variable	Pre-		-Post_0h_		χ^2^	*p*-Value	Pre-		-Post_24h_		χ^2^	*p*-Value	-Post_0h_		-Post_24h_		χ^2^	*p*-Value
	n	%	n	%			n	%	n	%			n	%	n	%		
pH ≤ 5	18	60.0%	13	43.3%	2.3	0.1	18	60.0%	12	40.0%	6.4	0.04	13	43.3%	12	40%	1.7	0.44
Proteinuria 1+ or higher	1	3.3%	13	43.3%	10.3	<0.01	1	3.3%	1	3.3%	0.0	1.0	13	43.3%	1	3.3%	10.3	<0.01
Glucosuria 1+ or higher	0	0.0%	0	0.0%	0.0	1.0	0	0.0%	0	0.0%	0.0	1.0	0	0.0%	0	0.0%	0.0	1.0
Hematuria 1+ or higher	1	3.3%	8	26.7%	12.2	<0.01	1	3.3%	3	6.7%	1.3	0.5	8	26.7%	3	6.7%	6.0	0.04
USG > 1.020	10	33.3%	20	66.7%	3.2	0.1	10	33.3%	14	46.7%	2.2	0.3	20	66.7%	14	46.7%	5.3	0.07

USG = urine specific gravity.

**Table 3 ijerph-18-10217-t003:** Perceived pain and physical exertion by measurement timepoints.

Variable	Pre-	1st Lap	2nd Lap	Post_0h_	*F* Value	*p*-Value	*ω_p_* ^2^
RPE	0.0 ± 0.0 (0.0 to 0.0)	6.4 ± 2.2 * (5.5 to 7.2)	7.6 ± 2.0 * (6.8 to 8.4)	8.8 ± 1.2 *^†¶^ (8.3 to 9.2)	176.4	<0.01	0.9 *large*
Lumbar Pain	0.0 ± 0.0 (0.0 to 0.0)	0.0 ± 0.0 (0.0 to 0.0)	1.0 ± 1.8 *^†^ (0.3 to 1.7)	0.9 ± 2.1 *^†^ (0.1 to 1.8)	4.2	<0.01	0.1 *moderate*
Squat Pain	0.1 ± 0.4 (0.08 to 0.2)	0.4 ± 1.1 (0.1 to 0.8)	2.7 ± 3.4 *^†^ (1.3 to 4.1)	2.5 ± 3.2 *^†^ (1.2 to 3.7)	11.7	<0.01	0.3 *large*
One-leg squat Pain	0.1 ± 0.4 (0.08 to 0.2)	0.3 ± 1.0 (0.1 to 0.7)	1.7 ± 2.8 * (0.5 to 2.8)	1.9 ± 3.0 *^†^ (0.7 to 3.2)	6.8	<0.01	0.2 *large*

Significant differences (*p* < 0.05) with * Pre-, ^†^ 1st Lap and ^¶^ 2nd Lap. RPE = rate or perceived exertion.

## Data Availability

No applicable.

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
