# Peer review of "Outpatient Assessment of Mechanical Load, Heat Strain and Dehydration as Causes of Transitional Acute Kidney Injury in Endurance Trail Runners"

_ijerph, 2021, doi:10.3390/ijerph181910217_

Round 1
Reviewer 1 Report
Dear Authors, The article “Outpatient assessment of mechanical workload, heat strain and 2 dehydration as causes of transitional acute kidney injury in en-3 durance trail runners ” is interesting, but several parts of the manuscript several parts of the manuscript could be added.
Specific comments:
Line 119: Where were temperature and humidity measured? At the launch site or is it the average temperature along the route (at all elevation levels)?
Can it be established what the performance level of runners was and what their level of training was? Numbers of training hours are given (line 102), but can their weekly or monthly training volume of kilometres run be specified as well? If so, was a link found between the performance level and changes in the monitored parameters or between the level of training and changes in the monitored parameters?
2.2 Study design (Line 114) Although monitoring the drinking regimen is listed among the limits of the work, it would be useful to add to the design whether, and if so at which moments (after each round or whenever?), refreshments were possible.
Author Response
R1.1 Dear Authors, The article “Outpatient assessment of mechanical workload, heat strain and 2 dehydration as causes of transitional acute kidney injury in en-3 durance trail runners ” is interesting, but several parts of the manuscript several parts of the manuscript could be added.
R/ We really appreciate the reviewer´s comments on the manuscript. This is an opportunity to improve the final version of the manuscript and we are very confident that the recommendations were very helpful in this sense. Authors want to thank reviewer 1 for his/her valuable time and dedication. Please see improvements in track change mode.
R1.2 Line 119: Where were temperature and humidity measured? At the launch site or is it the average temperature along the route (at all elevation levels)? R/We really appreciate the opportunity to clarify. Please see the correction of the text in the manuscript
R1.2. R/ We really appreciate the reviewer´s comments on the manuscript. This is an opportunity to improve the final version of the manuscript and we are very confident that the recommendations were very helpful in this sense. Authors want to thank reviewer 1 for his/her valuable time and dedication. Please see improvements in track change mode.
R1.3 Can it be established what the performance level of runners was and what their level of training was? Numbers of training hours are given (line 102), but can their weekly or monthly training volume of kilometres run be specified as well? If so, was a link found between the performance level and changes in the monitored parameters or between the level of training and changes in the monitored parameters?
R/ In this sense, the participants were asked to report trained hours per week considering that the amount of kilometers in trail running sometimes is not the best parameter to register total external load due to changes in ascend and descend total meters trained. We really try to select a homogenized sample so the level of the runner would not affect the results. We agree that this could be a future research topic, so we propose it in the corresponded section.
R1.4 Study design (Line 114) Although monitoring the drinking regimen is listed among the limits of the work, it would be useful to add to the design whether, and if so at which moments (after each round or whenever?), refreshments were possible.
R/ We agree that this information could be critical and it was clarify in the text in correction
R1.4. Please see the new section.
Reviewer 2 Report
The research topic is of interest, congratulations, but the manuscript requires major improvements, below examples:
All manuscript should be carefully reviewed assuming the IJERPH template and guidelines (affiliations post code, letter format, space between paragraphs, refs format, letter and paragraph size, and others).
Line 22 - Looks like more than one space before “perform”. Please confirm.
Numbers in abstracts before topics are unnecessary.
Line 65 - Looks more than one space before “[12]”- Please confirm.
Line 92 - “Due”.
Line 95 - The study aim should be more reasoned and directly and specifically indicated, considering athlete’s level, and running distance (the protocol).
99 - Please indicate how the athletes were selected (convenience in local club?) and the inclusion criteria.
In text and figure 1 “km” is in different format. Please standardize.
Please try to provide a better-quality figure 1 aiming readers interpretation.
138, 167 - Please indicate the cities in USA in full, as stated for other countries.
Please format tables 2 (example “category” and “528 ± 345.2 * (374.9 to 680.9)” - decimals and format). In the legend, please indicate what “*” and others represent.
258 - “table 2”?
Table 2 - “60.0%”. Please carefully review all table content.
Please consider table 3 similar organization comparing to table 1 and 2. Maybe the content will be better organized.
Figure 3 should be introduced with text, and the content organized according to journal guidelines. For example, the letter doesn’t seem to be Palatyno Linoty.
291 – “The main results” paragraph is suggested for readers better and faster understanding, previous to point by point analysis.
The “discussion subtopic” should be more developed and supported with references.
390 - AKI previously in full.
Please review “Author Contributions format”.
414-416 - Please check
Please carefully review the English throughout the manuscript.
Author Response
Reviewer: 2
R2.1 The research topic is of interest, congratulations, but the manuscript requires major improvements, below examples:
R/ We really appreciate the reviewer´s comments on the manuscript. This is an opportunity to improve the final version of the manuscript and we are very confident that the recommendations were very helpful in this sense. Authors want to thank reviewer 2 for his/her valuable time and dedication. Please see improvements in track change mode.
R2.2 All manuscript should be carefully reviewed assuming the IJERPH template and guidelines (affiliations post code, letter format, space between paragraphs, refs format, letter and paragraph size, and others).
R/We really appreciate the reviewer to pointing out this format issues. We considered the lastest IJERPH format, please see changes in the new manuscript version.
R2.3 Line 22 - Looks like more than one space before “perform”. Please confirm.
R/ Agreed, corrected.
R2.4 Numbers in abstracts before topics are unnecessary.
R/ Agreed, corrected.
R2.5 Line 65 - Looks more than one space before “[12]”- Please confirm.
R/ Agreed, corrected.
R2.6 Line 92 - “Due”.
R/ Agreed, corrected.
R2.7 Line 95 - The study aim should be more reasoned and directly and specifically indicated, considering athlete’s level, and running distance (the protocol).
R/ The main objective of the study was corrected following the reviewers recommendations.
R2.8 99 - Please indicate how the athletes were selected (convenience in local club?) and the inclusion criteria.
R/ We really appreciate the opportunity to clarify this issues. We added more information in this sense. Please see lines 103-108.
R2.9 In text and figure 1 “km” is in different format. Please standardize. R2.10 Please try to provide a better-quality figure 1 aiming readers interpretation.
R/ We have change the figure 1 considering the recommendations of the reviewer 2. Please see a better figure 1, considering quality and clarity.
R2.11. 138, 167 - Please indicate the cities in USA in full, as stated for other countries.
R/ City was added and USA country name was extended.
R2.12. Please format tables 2 (example “category” and “528 ± 345.2 * (374.9 to 680.9)” - decimals and format). In the legend, please indicate what “*” and others represent.
R/The corrections were made according to the recommended.
R2.13. 258 - “table 2”?
R/Corrected.
R2.14. Table 2 - “60.0%”. Please carefully review all table content.
R/Table 2 and 3 were corrected in this sense.
R2.15Please consider table 3 similar organization comparing to table 1 and 2. Maybe the content will be better organized.
R/It was not considered due to the amount of measurement moments represented in Table 3 compared to talbe 1 and 2.
R2.16. Figure 3 should be introduced with text, and the content organized according to journal guidelines. For example, the letter doesn’t seem to be Palatyno Linoty.
R/ The figures and tables were corrected to match journal format and typo.
R2.17. 291 – “The main results” paragraph is suggested for readers better and faster understanding, previous to point by point analysis.
R/ This new paragraph was added as requested.
R2.18. The “discussion subtopic” should be more developed and supported with references.
R/The discussion was rewritten in order to present better the results and support the analyses.
R2.19. 390 - AKI previously in full.
R/AKI was presented as suggested.
R2.20. Please review “Author Contributions format”.
R/ Reviewed
R2.21. 414-416 - Please check
R/ The conclusion section was reviewed and corrected according to the highlighted issues.
R2.22. Please carefully review the English throughout the manuscript.
Reviewer 3 Report
The manuscript reports a result of the relationship between mechanical workload, heat strain and dehydration as causes of transitional acute kidney injury in endurance trail runners. The results showed that there was a change in all serum measures as well as a correlation with mechanical workload. This manuscript needs to be more clear and concise with a more revealing command of future studies. The authors have tried to provide an informative and meaningful addition to the current study field, however, there are several changes that the authors are encouraged to revise to elevate the overall contribution of the paper to this research field. Importantly, I feel the level of English in the manuscript needs to be improved before resubmission to any journal. In addition, I would try to simplify and streamline the study for the reader. In my opinion, at the present moment, it reads as though there are so many variables that it's very difficult to follow throughout. I believe there are two main results - 1) serum/pain/exertion markers increase following endurance exercise, and 2) there is a significant correlation between serum indicators and mechanical workload. I would try to streamline this approach.
Abstract: - I would suggest deleting the (1), (2), etc...
Clean up English grammar - e.g., "They were assessed in serum test..." & "There was found a significant pattern of...."
post0hr - is this immediately post? Hard to read in abstract
Some p's italicised and some not - be consistent
AKI needs to be defined
Introduction:
What is cycling triathlon?
Be careful with placement of references - seems random?
English grammar - lines 41-42 - "and relative affordable and user-friendly require equipment to it practice. This disciplines...."
line 50 - "muscular affectation" - what does this mean?
lines 57-59 grammar - "This conditions is known as acute kidney injury (AKI) also called acute renal failure, an most of the times is reversible and asymptomatic [8,10]."
Although the third paragraph attempts to describe the most important/popular ways to quantify renal function, I feel as though the reader needs more context to the mentioned methods.
Line 92 - beginning the sentence with "due"
Overall, I feel as though the intro should get to AKI a little sooner and describe it's importance in endurance activity ASAP.
Methods:
This was a race?
Again, numerous English grammar errors throughout - e.g., line 179 - "in an isolated tend." - tent?
Could the liquid and food intake have influenced results?
Hard to read text in figure 1
"repeated measures t-test analysis." is this a paired samples t-test?
>0.2 - this means greater than 0.2
Results:
Where is figure 2?
Extremely difficult to follow and understand results
Why was BW and Htc not measured at the 24 hour post timepoint?
Discussion/Conclusion:
I feel as though you have so many variables to discuss that the discussion and conclusion is difficult to follow and does not follow a streamlined approach.
Author Response
Reviewer: 3
The manuscript reports a result of the relationship between mechanical workload, heat strain and dehydration as causes of transitional acute kidney injury in endurance trail runners. The results showed that there was a change in all serum measures as well as a correlation with mechanical workload. This manuscript needs to be more clear and concise with a more revealing command of future studies. The authors have tried to provide an informative and meaningful addition to the current study field, however, there are several changes that the authors are encouraged to revise to elevate the overall contribution of the paper to this research field. Importantly, I feel the level of English in the manuscript needs to be improved before resubmission to any journal. In addition, I would try to simplify and streamline the study for the reader. In my opinion, at the present moment, it reads as though there are so many variables that it's very difficult to follow throughout. I believe there are two main results - 1) serum/pain/exertion markers increase following endurance exercise, and 2) there is a significant correlation between serum indicators and mechanical workload. I would try to streamline this approach.
R/ We really appreciate the reviewer´s comments on the manuscript. This is an opportunity to improve the final version of the manuscript and we are very confident that the recommendations were very helpful in this sense. Authors want to thank reviewer 2 for his/her valuable time and dedication. Please see improvements in track change mode.
Abstract: - I would suggest deleting the (1), (2), etc...
R/ numbers were deleted as suggested.
Clean up English grammar - e.g., "They were assessed in serum test..." & "There was found a significant pattern of...."
R/ The sentences were corrected. Also a extensively grammar/language revision was made and rewritten as requested.
post0hr - is this immediately post? Hard to read in abstract
R/ This issue was clarified.
Some p's italicised and some not - be consistent
R/ This issue was reviewed and corrected throughout the article.
AKI needs to be defined
R/It was defined in abstract and reviewed throughout the manuscript.
Introduction:
What is cycling triathlon?
R/ The sentences should be understood as two different sports, it was corrected.
Be careful with placement of references - seems random?
R/ the manuscript was reviewed and corrected in this sense.
English grammar - lines 41-42 - "and relative affordable and user-friendly require equipment to it practice. This disciplines...."
R/ The manuscript was language and Grammarly reviewed.
line 50 - "muscular affectation" - what does this mean?
R/ Authors mean damage, it was corrected.
lines 57-59 grammar - "This conditions is known as acute kidney injury (AKI) also called acute renal failure, an most of the times is reversible and asymptomatic [8,10]."
R/The sentence was rewritten considering the mistakes.
Although the third paragraph attempts to describe the most important/popular ways to quantify renal function, I feel as though the reader needs more context to the mentioned methods.
R/ More context on the methods of AKI and muscle damage identification was given.
Line 92 - beginning the sentence with "due".
R/ this issue was corrected.
Overall, I feel as though the intro should get to AKI a little sooner and describe it's importance in endurance activity ASAP.
R/ The introduction was rewritten considering this recommendation.
Methods:
This was a race?
R/ This issue was clarified in the methods section.
Again, numerous English grammar errors throughout - e.g., line 179 - "in an isolated tend." - tent?
R/English grammar error were reviewed and corrected.
Could the liquid and food intake have influenced results?
R/This issue was addressed in the discussion methods. And the way the athletes performed hydration and food intake was also described in methods sections.
Hard to read text in figure 1
R/ figure 1 was corrected and clarified. A new Figure 1 was added to the manuscript.
"repeated measures t-test analysis." is this a paired samples t-test?
R/ we really appreciate the reviewer for highlight this issue, it was clarified.
>0.2 - this means greater than 0.2
R/ we want to thank the reviewer for identify this mistake. It was also corrected.
Results:
Where is figure 2?
R/ The number of figures were corrected and the figure 2 was added to the text to better clarify the results.
Extremely difficult to follow and understand results
R/ The results section was rewritten as suggested.
Why was BW and Htc not measured at the 24 hour post timepoint?
R/ This was due to logistics of the study, could be understood as a study limitation but it was due to some complications.
Discussion/Conclusion:
I feel as though you have so many variables to discuss that the discussion and conclusion is difficult to follow and does not follow a streamlined approach.
R/The discussion sections was rewritten in order to clarify, considering the abovementioned issues.
Round 2
Reviewer 2 Report
The manuscript improved. Congratulations.
Although some improvements should still be performed, below described.
7-17 Please provide all institutions zip code and authors abbreviations.
46 - Please review and follow journal template - Subtopics format, no space between paragraphs, and others.
113 - The text indicates what was performed. But what was the aim/objective? Please consider reformulate for readers comprehension.
174 - Journal guidelines “Cockcroft and Gault” not“Cockcroft & Gault”.
230 vs. 238 “30 min” vs. “5min”. Please standardize these format examples in all manuscript (another examples 253 “km” with space and previously without space).
243 - “Player Load” and “Impacts” should be described, namely how they are determined and what represent.
245 - “rate of perceived exertion”. Should be abbreviated herein. In line 251 only “RPE”.
301 - Please carefully review all tables and figures content. For example, In Figure 2 after 3.3 “%” is missing and some cases present “%” immediately after the value and in others not. Some values with one decimal and others not, standardization suggested.
303-305 / 331-334 – Legend. Please confirm journal template line space.
342 - “40.0”? The next value (“23.4”) presents one decimal. Standardization suggested regarding decimals.
468 - “Conceptualization - Please correct.
475 - End point missing.
488 - Please double check the references format, they should be according to the journal guidelines.
Please double check the English language throughout all the manuscript.
Author Response
Dear Editor and reviewers:
We have carefully considered all reviewers' recommendations for the paper (ijerph-1332420) entitled "Outpatient assessment of mechanical workload, heat strain and dehydration as causes of transitional acute kidney injury in endurance trail runners”. Please find enclosed our detailed answers to reviewers' queries. The authors declare that the manuscript is original and has not been considered for publication elsewhere. Additionally, the authors had approved the paper for release and are in agreement with its content.
The manuscript improved. Congratulations.
R/ We want to give our appreciation for the reviewers comments and recommendations, which significantly improve the final outcome.
Although some improvements should still be performed, below described.
R/ We have considered all the requirements.
7-17 Please provide all institutions zip code and authors abbreviations.
R/The information was added to each author´s affilliation
46 - Please review and follow journal template - Subtopics format, no space between paragraphs, and others.
R/All format issues were solved considering IJERPH template
113 - The text indicates what was performed. But what was the aim/objective? Please consider reformulate for readers comprehension.
R/ The aim of the study was clarified as suggested.
174 - Journal guidelines “Cockcroft and Gault” not“Cockcroft & Gault”.
R/ Citations was corrected as requested
230 vs. 238 “30 min” vs. “5min”. Please standardize these format examples in all manuscript (another examples 253 “km” with space and previously without space).
R/ we really appreciate for identify this mistakes, we have solved it all.
243 - “Player Load” and “Impacts” should be described, namely how they are determined and what represent.
R/ Player Load and Impact variables were defined in the methods section.
245 - “rate of perceived exertion”. Should be abbreviated herein. In line 251 only “RPE”.
R/ The abbreviation was corrected.
301 - Please carefully review all tables and figures content. For example, In Figure 2 after 3.3 “%” is missing and some cases present “%” immediately after the value and in others not. Some values with one decimal and others not, standardization suggested.
R/ All figures and tables were reviewed and corrected based on this recommendation.
303-305 / 331-334 – Legend. Please confirm journal template line space.
R/The format was corrected based on IJERPH template
342 - “40.0”? The next value (“23.4”) presents one decimal. Standardization suggested regarding decimals.
R/All data was reviewed and corrected accordingly, one decimal was used throughout the text.
468 - “Conceptualization - Please correct.
R/Corrected
475 - End point missing.
R/End point added.
488 - Please double check the references format, they should be according to the journal guidelines.
R/The references were checked according to the journal guidelines.
Please double check the English language throughout all the manuscript.
R/English grammar and language was corrected by a native speaker.
Reviewer 3 Report
Thank you for your resubmission. I still believe the manuscript should be read and edited by a native English speaker.
Here are some edits I recommend:
line 22/24 - be consistent with distance 35 or 35.3 km?
line 25 - moments or timepoints?
line 26 - do you need to define post24hr?
line 33 - this change correlate with - grammar
lines 49-50 - doesn't make sense - grammar
lines 50-51 - "These disciplines have been found to push the body limits in multiple factors" - what does this mean?
lines 58-59 - "it is common to present muscular damage"?
lines 85-86 - should there be a reference here? - "Recent research has suggested that high physical load is the common denominator of AKI development."
"and among >18 years old," - unsure of the meaning here?
line 136 - moments
lines 265-266 - revisit as the numbers don't add up (e.g., what would 0.80 be classified as? - "< 0.2 trivial; 0.2-0.49 small; 0.5-0.79 moderate and >0.8 large [24] when corresponded."
I still find the results and discussion hard to follow
Author Response
Dear Editor and reviewers:
We have carefully considered all reviewers' recommendations for the paper (ijerph-1332420) entitled "Outpatient assessment of mechanical workload, heat strain and dehydration as causes of transitional acute kidney injury in endurance trail runners”. Please find enclosed our detailed answers to reviewers' queries. The authors declare that the manuscript is original and has not been considered for publication elsewhere. Additionally, the authors had approved the paper for release and are in agreement with its content.
Please find all corrections in red inside the manuscript.
Thank you for your resubmission. I still believe the manuscript should be read and edited by a native English speaker.
R/English grammar and language was corrected by a native speaker.
Here are some edits I recommend:
line 22/24 - be consistent with distance 35 or 35.3 km?
R/The text was reviewed in relation to the consistency of the km, min, and other values considering this commentary.
line 25 - moments or timepoints?
R/The word moments was changed for timepoints throughout the manuscript as suggested.
line 26 - do you need to define post24hr?
R/Post 24 h was defined as suggested
line 33 - this change correlate with – grammar
R/The sentence was change for clarity
lines 49-50 - doesn't make sense – grammar
R/ The sentence was changed
lines 50-51 - "These disciplines have been found to push the body limits in multiple factors" - what does this mean?
R/ The sentence was change for clarity
lines 58-59 - "it is common to present muscular damage"?
R/A citation was added to the statement.
lines 85-86 - should there be a reference here? - "Recent research has suggested that high physical load is the common denominator of AKI development."
R/A citation was added to the statement.
"and among >18 years old," - unsure of the meaning here?
R/The sentence was clarified accordingly.
line 136 – moments
R/The word moments, was changed for timepoints as suggested
lines 265-266 - revisit as the numbers don't add up (e.g., what would 0.80 be classified as? - "< 0.2 trivial; 0.2-0.49 small; 0.5-0.79 moderate and >0.8 large [24] when corresponded."
R/The qualification of the magnitude of change was added to each value in table 1 and 3
I still find the results and discussion hard to follow
R/The results and discussion were corrected as suggested for more clarity.